# The role of ecological niche evolution on diversification patterns of birds distinctly distributed between the Amazonia and Atlantic rainforests

Ricardo Ribeiro da Silva[1,2]* , Bruno Vilela[3] , Daniel Paiva Silva[4] , André Felipe Alves de Andrade[5] , Pablo Vieira Cerqueira[1,6] , Gabriela Silva Ribeiro Gonçalves[1,6] , Marcos Pérsio Dantas Santos[1]

1 ICB—BIOMACRO-Lab—Laboratório de Biogeografia da Conservação e Macroecologia, Universidade Federal do Pará, Belém, PA, Brazil, 2 Sociedade para a Conservação da Aves do Brasil, Rua Vereador José Cícero Epifânio, Lagoa dos Gatos, PE, Brazil, 3 Instituto de Biologia, Universidade Federal da Bahia, Salvador, Bahia, Brazil, 4 COBIMA Lab, Departamento de Ciências Biológicas, Rodovia Geraldo Silva Nascimento, Instituto Federal Goiano, Urutaí, Goiás, Brazil, 5 Theory, Metapopulation, and Landscape Ecology Lab, Programa de Pós-Graduação em Ecologia e Evolução, Instituto de Ciências Biológicas, Universidade Federal de Goiás, Goiânia, GO, Brazil, 6 Museu Paraense Emílio Goeldi, Curso de Pós-Graduação de Zoologia, Universidade Federal do Pará, Belém, Pará, Brazil

☯ These authors contributed equally to this work.
* ricardo.ornitologo@gmail.com

**Data Availability Statement:** All relevant data are within the paper and its Supporting Information files.

## Abstract

The Amazonian and Atlantic Forest share several organisms that are currently isolated but were continuously distributed during the Quaternary period. As both biomes are under different climatic regimes, paleoclimatic events may have modulated species' niches due to a lack of gene flow and imposing divergent selection pressure. Here, we assessed patterns of ecological niche overlap in 37 species of birds with disjunct ranges between the Amazonian and Brazilian Atlantic Forests. We performed niche overlap analysis and ecological niche modeling using four machine-learning algorithms to evaluate whether species' ecological niches evolved or remained conserved after the past South American biogeographic events. We found a low niche overlap among the same species populations in the two biomes. However, niche similarity tests showed that, for half of the species, the overlap was higher than the ones generated by our null models. These results lead us to conclude that niche conservatism was not enough to avoid ecological differentiation among species even though detected in many species. In sum, our results support the role of climatic changes in late-Pleistocene—that isolated Amazon and the Atlantic Forest—as a driving force of ecological differences among the same species populations and potential mechanism of current diversification in both regions.

## Introduction

The Quaternary paleoclimatic cycles of glaciation and interglaciation events profoundly affected neotropical rainforests ecosystems [1–6]. During the Last Glacial Maximum (LGM;

**Funding:** MPDS and DPS were supported by CNPq research productivity fellowships (Proc. Number: 308403/2017-7 and 304494/2019-4, respectively). PVC was supported by a CAPES (the Brazilian Higher Education Training Program) doctoral fellowship (# 1537057). RRDS was supported by a CAPES master's scholarship (# 1666680). Federal University of Pará (UFPA) support the payment of publication fees (PROPESP-PAPQ 01/2020 - QUALIFIED PUBLICATION SUPPORT PROGRAM).

**Competing interests:** The authors have declared that no competing interests exist.

21,000 YBP), low temperatures, dry climates, and the low $CO_2$ concentration favored the expansion of C4 plants [7]. As a consequence, neotropical savannas expanded replacing forested areas thus forming the South American "Dry Diagonal" isolating the biota of the two largest rainforests of South America: the Amazon and the Atlantic Forest [8].

The retraction of the historical connections (at least three of them are well-supported: [4, 9, 10], at the end of the Pleistocene, isolated populations of the same species that occurred in both biomes [6]. This isolation prevented gene flow, keeping disjunct populations under different evolutionary pressures that might ultimately driven allopatric speciation [11]. Some authors argued that the time since the Last Glacial Maximum might not have been enough to generate speciation, especially for species with slow life cycles [12–15]. However, the isolation of the two forests may have led to observable ecological differentiation that could set the path for speciation in the long run. For instance, pairs of disjunct sister species of bees (with nearly indistinguishable morphologic characteristics) that occur in the Amazon and the Atlantic Forest had lower ecological niche overlap compared to pairs of phylogenetically distant species in the same biome [16]. This result could indicate that the rapid Grinnellian niche (i.e., the set of coarse-grained environmental conditions suitable for the persistence of a species, *sensu* [17]) evolution resulted from the populations' rupture after the two forests separated utterly. Still, there are a handful of examples that may be cited with plants [18, 19], birds [4, 20], mammals [21, 22], and amphibian species [9, 23].

Even though there is support for a fast Grinnellian niche differentiation, a larger body of evidence shows that species do tend to retain their ancestral niche—the so-called niche conservatism [24]. Previous results suggest that ecological differentiation commonly emerges in deep evolutionary time (taxonomic family level), much more than the last 10 thousand years since the last glacial maximum took place [24, 25]. Such evidence also suggests that speciation would begin before considerable changes in the ecological niche being noticeable. This phenomenon would happen because ecological niches are the product of a complex combination of physiological, morphological, behavioral, and ecological traits that are under several evolutionary constraints [26]. Hence, multiple trait changes may be necessary before they can reflect into ecological niche evolution. Different from the bees in the example mentioned above, vertebrate species have a slower life cycle. Therefore, the time since the last glaciation might not have been enough for populations to have evolved different ecological niches.

Investigating how fast vertebrate species can modify their ecological niches is central to understand the speciation process and ongoing diversification drivers in the two most diverse tropical forests of South America. In this paper, we used 37 bird species disjunctly distributed between Amazon and Atlantic Forest to test whether their populations show signs of rapid Grinnellian niche evolution. For this, we used two approaches: (I) we first tested whether ecological niche models built from occurrence records of one population could predict the distribution of the other; (II) we also examined the niche overlap between populations using a recently developed framework, which considers the differences in environmental conditions spatially available in each region—such differences, if not taken into account, could mask actual niche evolution [27].

## Methods

### Target species and database

The 37 selected bird species (Table 1) have unique disjunct distribution between the Amazon and the Atlantic Forest (Fig 1). Although there is a large number of species that present such distribution patterns, some of them occur in the Cerrado and Caatinga biomes, and this would

**Table 1. Bird species that are disjunctly distributed between Amazon and the Atlantic Forest, common names and, number of unique geographical records.**

| Order | Family | Taxon and author | English names | Records |
|---|---|---|---|---|
| Tinamiformes | Tinamidae | *Crypturellus variegatus* (Gmelin, 1789) | Variegated Tinamou | 476 |
| Nyctibiiformes | Nyctibiidae | *Nyctibius aethereus* (Wied, 1820) | Long-tailed Potoo | 168 |
| Caprimulgiformes | Caprimulgidae | *Antrostomus sericocaudatus* Cassin, 1849 | Silky-tailed Nightjar | 119 |
| | | *Nyctiphrynus ocellatus* (Tschudi, 1844) | Ocellated Poorwill | 289 |
| Apodiformes | Apodidae | *Panyptila cayennensis* (Gmelin, 1789) | Lesser Swallow-tailed Swift | 928 |
| | Trochilidae | *Lophornis chalybeus* (Temminck, 1821) | Festive Coquette | 184 |
| | | *Hylocharis sapphirina* (Gmelin, 1788) | Rufous-throated Sapphire | 246 |
| | | *Hylocharis cyanus* (Vieillot, 1818) | White-chinned Sapphire | 631 |
| | | *Discosura langsdorffi* (Temminck, 1821) | Black-bellied Thorntail | 115 |
| | | *Discosura longicaudus* (Gmelin, 1788) | Racket-tailed Coquette | 73 |
| | | *Heliothryx auritus* (Gmelin, 1788) | Black-eared Fairy | 557 |
| Trogoniformes | Trogonidae | *Trogon collaris* Vieillot, 1817 | Collared Trogon | 1114 |
| Galbuliformes | Bucconidae | *Monasa morphoeus* (Hahn & Küster, 1823) | White-fronted Nunbird | 769 |
| Piciformes | Picidae | *Picumnus exilis* (Lichtenstein, 1823) | Bahia Piculet | 279 |
| | | *Piculus flavigula* (Boddaert, 1783) | Yellow-throated Woodpecker | 606 |
| | | *Celeus torquatus* (Boddaert, 1783) | Ringed Woodpecker | 388 |
| Psittaciformes | Psittacidae | *Amazona farinosa* (Boddaert, 1783) | Mealy Parrot | 1925 |
| Passeriformes | Thamnophilidae | *Herpsilochmus rufimarginatus* (Temminck, 1822) | Rufous-winged Antwren | 754 |
| | | *Thamnomanes caesius* (Temminck, 1820) | Cinereous Antshrike | 623 |
| | | *Thamnophilus palliatus* (Lichtenstein, 1823) | Chestnut-backed Antshrike | 421 |
| | | *Cercomacroides laeta* (Todd, 1920) | Willis's Antbird | 64 |
| | Scleruridae | *Sclerurus caudacutus* (Vieillot, 1816) | Black-tailed Leaftosser | 186 |
| | Dendrocolaptidae | *Glyphorynchus spirurus* (Vieillot, 1819) | Wedge-billed Woodcreeper | 1816 |
| | Xenopidae | *Xenops minutus* (Sparrman, 1788) | Plain Xenops | 1801 |
| | Pipridae | *Ceratopipra rubrocapilla* (Temminck, 1821) | Red-headed Manakin | 438 |
| | | *Dixiphia pipra* (Linnaeus, 1758) | White-crowned Manakin | 577 |
| | Pipritidae | *Piprites chloris* (Temminck, 1822) | Wing-barred Piprites | 711 |
| | Rhynchocyclidae | *Mionectes oleagineus* (Lichtenstein, 1823) | Ochre-bellied Flycatcher | 1799 |
| | | *Rhynchocyclus olivaceus* (Temminck, 1820) | Olivaceous Flatbill | 488 |
| | | *Tolmomyias poliocephalus* (Taczanowski, 1884) | Gray-crowned Flycatcher | 782 |
| | | *Hemitriccus griseipectus* (Snethlage, 1907) | White-bellied Tody-Tyrant | 183 |
| | Tyrannidae | *Ornithion inerme* Hartlaub, 1853 | White-lored Tyrannulet | 554 |
| | | *Rhytipterna simplex* (Lichtenstein, 1823) | Grayish Mourner | 816 |
| | Vireonidae | *Hylophilus thoracicus* Temminck, 1822 | Lemon-chested Greenlet | 388 |
| | Turdidae | *Turdus fumigatus* Lichtenstein, 1823 | Cocoa Thrush | 360 |
| | Thraupidae | *Hemithraupis flavicollis* (Vieillot, 1818) | Yellow-backed Tanager | 596 |
| | Cardinalidae | *Habia rubica* (Vieillot, 1817) | Red-crowned Ant-Tanager | 1094 |

not allow us to classify them, *a priori*, with exclusively Amazonian or Atlantic populations. Thus, we excluded those species from our study.

We built our database from occurrence records available at the Global Biodiversity Information Facility (http://www.gbif.org), eBird (http://ebird.org/content/ebird/), Museu Paraense Emílio Goeldi, VertNet (http://vertnet.org/) and, WikiAves (http://www.wikiaves.com/). For each species, we compiled information on geographical coordinates, country, state, municipality, biome, genus, epithet, scientific name, sources, and identification number (voucher). We excluded species with less than ten geographically unique occurrence records in each biome, considering the resolution of the climatic variables (see below). We drew a minimum convex

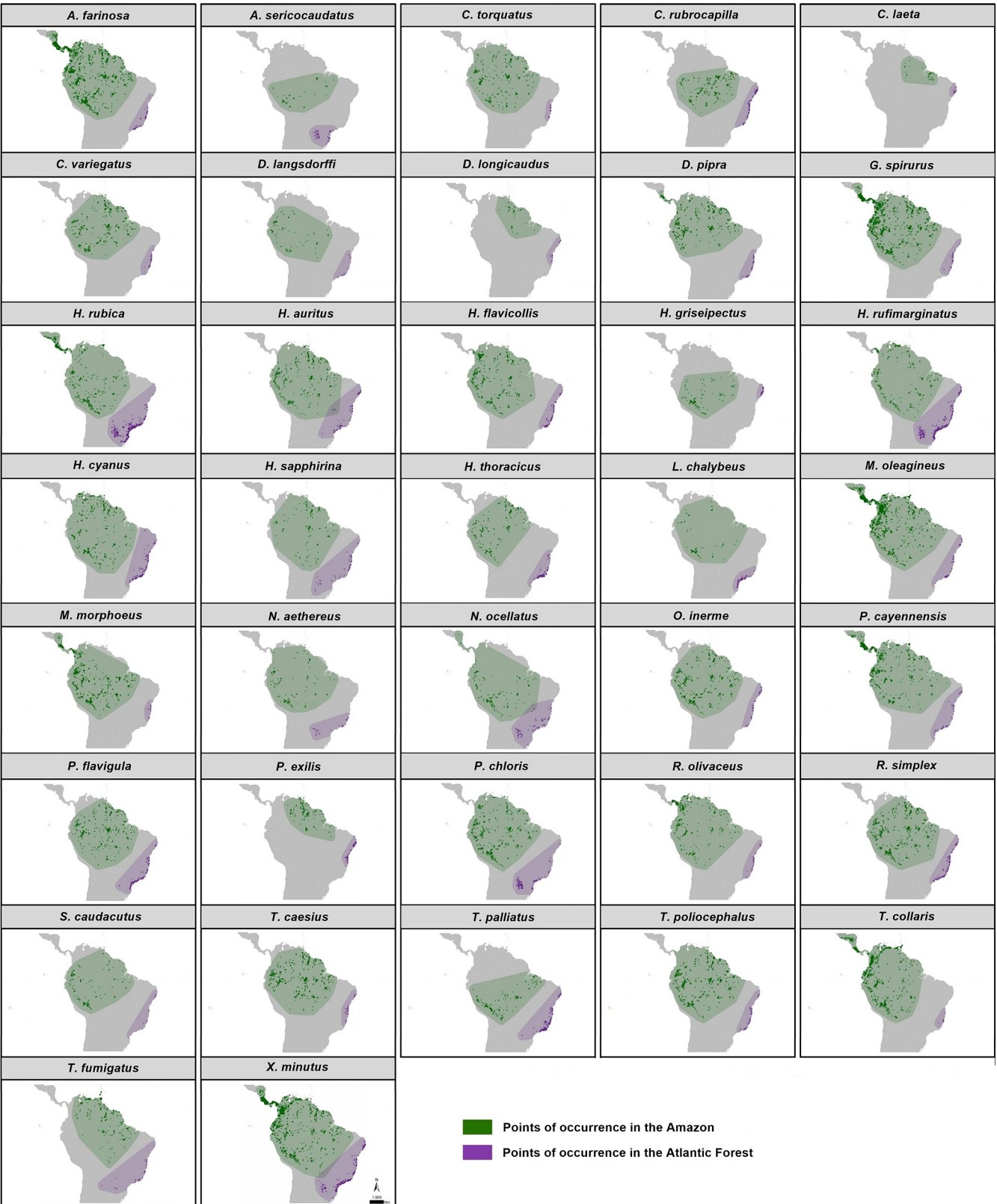

**Fig 1. Distribution of Amazonian and Atlantic Forest population of each species.** Green dots represent occurrence records of populations in the Amazon while purple ones represent population occurrences in the Atlantic Forest. Both green and purple areas were generated from the minimum convex polygon and a 1-degree buffer around each occurrence record for each species in both biomes.

polygon around the occurrences and added a 1° buffer (~111.19 km). We used Ornithological gazetteers [28–35] and Google Earth version 7.3 (2018) to obtain information on occurrence records that lacked geographical coordinates but had some information about the sampling site as long as they were registered by acknowledged observers (Conservation Units, institutes, and private properties). Records referencing only to the municipality were not included in the analysis. We obtained 23,318 unique occurrences for the 37 species (Table 1) from both Amazonian and Atlantic forest populations.

## Climate data

We downloaded all 19 environmental variables available at WorldClim 1.0 (http://www.worldclim.org/; [36]; see supplementary material) at a spatial resolution of 2.5 minutes (~16 km$^2$ area at the equator) for the entire Neotropical region. We calculated mean and standard deviation for each bioclimatic variable in grid cell values. Then, we subtracted mean value and the cell value and divided by standard deviation.

## Ecological niche modeling

We performed principal component analysis (PCA) and used a correlation matrix for the derivation of 19 components (PC) from which we retained the first seven axes as our final environmental layer (97% of all original climate variation, see supplementary material). The seven axes of the PCA were used as new predictor variables in the distribution models and were standardized to have zero mean and variance |1|. We use the environmental layers generated by PCA to avoid over-parameterization of the models due to the number of climatic variables [37–39], thus maximizing the variance explained by each component avoiding the correlation between variables [40].

For model construction, we partitioned species' occurrences according to their origin: Amazon or Atlantic Forest. We then used the data from one region to fit the model and evaluated its accuracy using occurrence data from the other biome. We performed this analysis on both ways, in which models fitted using occurrences from the Amazon were evaluated with data from the Atlantic Forest. Likewise, models fitted with Atlantic forest data were evaluated using data from the Amazon. By adjusting the model for one population and evaluating its effectiveness when predicting the other population (reciprocal analysis), we obtain an estimate of niche change, since these models characterize the niche of the species. Since we geographically partitioned species' data, there is a risk of artificially creating a sampling bias if pseudo-absence and background allocation are also not restricted [41]. In order to avoid this bias, we delimited the accessible area for each subset by calculating the minimum convex polygon and adjusting the buffer around the polygon [42]. The pseudo-absences selection method was used through a simple bioclimatic presence model—Bioclim [43, 44]—to randomly allocate the pseudo-absences with the environmental profile and accessible areas to the species along with the same number of presences for each species. Pseudo-absences were randomly sampled with the environmental profile (RSEP; [45]) restricted to species' accessible areas [46].

We built niche models using the following machine-learning algorithms, based on presence and pseudo-absences: *Support Vector Machine* (SVM), *Gaussian* (GAU), and based on Presence and Background, *Random Forest* (RDF) and, *General Linear Models* (GLM): i). The SVM method belongs to the family of generalized linear classifiers. This method obtains a limit

around the database, rather than estimating probability density [47], characterized by minimizing the probability of classifying patterns not observed by the probability distribution of the data erroneously [48]. The GLM method works with logistic regression, based on the relationship between the mean response variable and the linear combination of explanatory variables, suitable for ecological relationships analysis when dealing with non-linear data structures [49]. The GAU method is a probabilistic model in which Gaussian distribution models both the spatial dependency and the binary observation, generating the corresponding Gaussian variable [50]. The RDF method is an effective tool in forecasting, which uses individual predictors and correlations [51]. The RDF method builds several classification trees relating presences and absences to the environmental variables, and then combine all predictions based on trees frequency of classification [52]. The modeling process was carried out using the ENMTML package [53].

We used the *Receiver Operating Characteristic* (ROC) threshold, which balances commission and omission errors (sensitivity and specificity), to transform the suitability matrices into absence-presence maps. The area below the ROC threshold is known as AUC and serves as an evaluation measure of the model independently from the chosen limit [54]. The AUC values range from 0 to 1, with values below 0.5 indicating that the model has no better efficacy than a randomized distribution; values between 0.5–0.7 indicate acceptable accuracy while values between 0.7–0.9 indicate good accuracy. Lastly, the values above 0.9 indicate optimal predictions [55]. This procedure has criticisms about its use because it omits information about the goodness of model performance, the uncertainty of false positives, and their spatial distribution dimensions [56–58]. However, we used this evaluation method for providing results that optimize the probability thresholds by maximizing the percentage of actual presence and absences [56], as well as being widely used in niche modeling studies [48, 59–63]. Finally, to mitigate the errors and uncertainties of individual models, we used the ensemble technique, which consists of averaging out the best models for each species to generate a more robust prediction [64].

### Analysis of the species' climatic niche

We used the 19 bioclimatic variables obtained from WorlClim to compare the niche of 37 species that have populations in both Amazon and the Atlantic Forest. We performed a calibrated PCA in the entire environmental space of the study ([67], PCA-env) to measure the niche overlap of target species based on the use of the available environment. This method quantifies the overlap of climatic niche involving three steps: (1) calculation of the density of occurrence points and environmental factors along the environmental axes, using a multivariate analysis, (2) measurement of niche overlap over gradients of the multivariate analysis and (3) analysis of niche similarity through statistical tests [27].

We then calculated the niche overlap between the Amazonian and Atlantic populations using Schoener's D metric (1970). This metric ranges from 0 (without overlap) to 1 (complete overlap) for each pair of disjunctly distributed species [27]. Then, to evaluate the hypothesis of niche evolution or conservation, the method performs two different routines of randomization [27], through two distinct components in the niche comparison: the equivalence test and the similarity test. The niche equivalence test verifies whether the niche overlap between populations is constant by randomly relocating the occurrences of each species of both biomes [27]. The similarity test assesses whether related species occupy similarly, but rarely identical, niches [65]. In this test, we verify whether niche overlap values remain unchanged (1 to 2), followed by a reciprocal comparison (2 to 1) given a randomly distributed interval. We performed this test 1,000 times, which guarantees the rejection of the null hypothesis. However, if the

Schoener's D ranges within 95% of the simulated density values, the null hypothesis—niche equivalence—cannot be rejected [27].

Among the distinct components in the niche comparison, in addition to the niche equivalence and similarity tests, it is also possible to assess niche *stability*, *expansion* and *unfilling*, when comparing the known distributions of the species in both biomes. The stability test represents the niche proportion of species populations in a biome superimposing the niche in the other biome, demonstrating the stability that species retain their niches in space and time [66]. The expansion test evaluates the niche proportion of species populations in one biome that does not overlap with the niche of those in another biome [66]. In expansion, species occupy areas with different climatic conditions than those in the compared niche [67]. The unfilling test evaluates niche expansion of the populations in climatically analogous areas, only partially filling the climatic niche in this area, not overlapping with the compared niche [68]. We considered threshold values above 0.7 as high (results representing at least 70% of the analogous niche), between 0.5 and 0.7 as partial, and below 0.5 as low (results representing at most 50% of the analogous niche), for evaluation of the stability, expansion and unfilling test results based on previous studies [54, 69–74]. Finally, we estimated how much the niche of each species evolved or remained conserved in the Amazonian and Atlantic populations.

## Results

### Ecological niche models (ENM) of birds with disjunct distribution between Amazon and the Atlantic Forest

The ecological niche models of Amazonian populations presented higher values of AUC than those found for populations of the same species in the Atlantic Forest (Table 2). Such better performance was likely partially due to the higher number of occurrence records in the Amazon (n = 19,828, mean = 535.89) than in the Atlantic Forest (n = 3,490, mean = 94.32); which allowed a better characterization of the realized niche of Amazonian populations. Ecological niche models of Amazonian populations presented 57% (n = 21 species) of model with AUC values between 0.7 to 0.9 (considered a good performance), and 35% (n = 13) above 0.9 (considered optimal performance). Only 8% (n = 3) had AUC values between 0.5 and 0.7 (considered acceptable). For the Atlantic Forest ENMs, 73% (n = 27) had an acceptable prediction (AUC = 0.5–0.7), and 27% (n = 10) showed a good one (AUC = 0.7–0.9). The resultant potential distribution included both large and restricted distribution as in *Antrostomus sericocaudatus* Cassin, 1849, which is distributed from Southern Central America to Southern Uruguay and *Discosura longicaudus* (Gmelin, 1788) that is restricted to the north of the Atlantic Forest and North and Eastern Amazon.

In general, models of Amazonian populations were able to predict suitable areas in Atlantic Forest for 36 species: 49% (n = 18) of them predicted suitable areas for the populations in both north and south of the Atlantic Forest; 38% only in the northern part of the Atlantic Forest (n = 14); and 11% (n = 4) predicted suitable areas exclusively in southern Atlantic Forest (Fig 2). Only for *Monasa morpheus* (Hahn & Küster, 1823), the model was unable to predict distribution beyond Amazon.

Models of Atlantic Forest populations predicted suitable areas in Amazon for all 37 species analyzed. Western and Eastern Amazon regions were present in 14% (n = 5) of the predicted areas. The predicted area mostly covered Western Amazon (76%; n = 28), and less frequently, it included the eastern region of this biome (11%; n = 4). Predictions in the Central-South Region of the Cerrado appeared in 22% (n = 8) of the Amazonian models and 38% (n = 14) of Atlantic models.

**Table 2. Modeling results we obtained for each one of the models species we evaluated.**

| Taxon | ENS | | RDF | | SVM | | GAU | | GLM | |
|---|---|---|---|---|---|---|---|---|---|---|
| | **AM** | **MA** | **AM** | **MA** | **AM** | **MA** | **AM** | **MA** | **AM** | **MA** |
| *Amazona farinosa* | 0.898 | 0.77 | 0.971 | 0.763 | 0.872 | 0.736 | 0.878 | 0.764 | 0.87 | 0.816 |
| *Antrostomus sericocaudatus* | 0.667 | 0.618 | 0.371 | 0.651 | 0.729 | 0.573 | 0.604 | 0.582 | 0.427 | 0.665 |
| *Celeus torquatus* | 0.889 | 0.594 | 0.964 | 0.61 | 0.795 | 0.41 | 0.906 | 0.579 | 0.892 | 0.368 |
| *Ceratopipra rubrocapilla* | 0.827 | 0.641 | 0.776 | 0.519 | 0.825 | 0.711 | 0.844 | 0.618 | 0.863 | 0.718 |
| *Cercomacroides laeta* | 0.637 | 0.551 | 0.529 | 0.362 | 0.533 | 0.521 | 0.433 | 0.582 | 0.848 | 0.287 |
| *Crypturellus variegatus* | 0.934 | 0.645 | 0.981 | 0.638 | 0.901 | 0.685 | 0.926 | 0.613 | 0.927 | 0.487 |
| *Discosura langsdorffi* | 0.901 | 0.679 | 0.91 | 0.713 | 0.903 | 0.673 | 0.861 | 0.651 | 0.931 | 0.446 |
| *Discosura longicaudus* | 0.849 | 0.663 | 0.479 | 0.809 | 0.878 | 0.261 | 0.889 | 0.529 | 0.778 | 0.65 |
| *Dixiphia pipra* | 0.87 | 0.615 | 0.985 | 0.681 | 0.891 | 0.587 | 0.769 | 0.586 | 0.837 | 0.607 |
| *Glyphorhynchus spirurus* | 0.971 | 0.762 | 0.988 | 0.788 | 0.941 | 0.699 | 0.972 | 0.722 | 0.982 | 0.838 |
| *Habia rubica* | 0.907 | 0.763 | 0.926 | 0.757 | 0.908 | 0.776 | 0.944 | 0.754 | 0.85 | 0.764 |
| *Heliothryx auritus* | 0.903 | 0.671 | 0.903 | 0.684 | 0.924 | 0.734 | 0.94 | 0.725 | 0.845 | 0.541 |
| *Hemithraupis flavicollis* | 0.883 | 0.635 | 0.976 | 0.615 | 0.931 | 0.648 | 0.853 | 0.641 | 0.771 | 0.478 |
| *Hemitriccus griseipectus* | 0.843 | 0.637 | 0.874 | 0.637 | 0.884 | 0.451 | 0.85 | 0.484 | 0.763 | 0.467 |
| *Herpsilochmus rufimarginatus* | 0.83 | 0.642 | 0.871 | 0.568 | 0.805 | 0.694 | 0.852 | 0.679 | 0.791 | 0.629 |
| *Hylocharis cyanus* | 0.911 | 0.659 | 0.936 | 0.581 | 0.962 | 0.689 | 0.938 | 0.658 | 0.807 | 0.708 |
| *Hylocharis sapphirina* | 0.899 | 0.753 | 0.947 | 0.753 | 0.909 | 0.773 | 0.91 | 0.805 | 0.829 | 0.682 |
| *Hylophilus thoracicus* | 0.802 | 0.681 | 0.939 | 0.731 | 0.91 | 0.718 | 0.795 | 0.595 | 0.565 | 0.418 |
| *Lophonis chalybeus* | 0.749 | 0.576 | 0.472 | 0.548 | 0.731 | 0.578 | 0.763 | 0.593 | 0.754 | 0.585 |
| *Mionectes oleagineus* | 0.987 | 0.715 | 0.987 | 0.76 | 0.981 | 0.703 | 0.994 | 0.726 | 0.987 | 0.672 |
| *Monasa morphoeus* | 0.97 | 0.639 | 1,000 | 0.719 | 0.997 | 0.674 | 0.924 | 0.663 | 0.959 | 0.502 |
| *Nyctibius aethereus* | 0.833 | 0.583 | 0.704 | 0.55 | 0.93 | 0.56 | 0.871 | 0.596 | 0.826 | 0.624 |
| *Nyctiphrynus ocellatus* | 0.872 | 0.719 | 0.917 | 0.744 | 0.949 | 0.655 | 0.878 | 0.665 | 0.746 | 0.815 |
| *Ornithion inerme* | 0.868 | 0.673 | 0.931 | 0.74 | 0.902 | 0.609 | 0.811 | 0.634 | 0.829 | 0.709 |
| *Panyptila cayennensis* | 0.918 | 0.748 | 0.886 | 0.78 | 0.885 | 0.701 | 0.966 | 0.759 | 0.935 | 0.753 |
| *Piculus flavigula* | 0.724 | 0.667 | 0.834 | 0.582 | 0.741 | 0.742 | 0.664 | 0.75 | 0.656 | 0.593 |
| *Picumnus exilis* | 0.757 | 0.648 | 0.708 | 0.663 | 0.663 | 0.39 | 0.755 | 0.597 | 0.904 | 0.684 |
| *Piprites chloris* | 0.868 | 0.686 | 0.916 | 0.702 | 0.898 | 0.683 | 0.893 | 0.675 | 0.766 | 0.475 |
| *Rhynchocyclus olivaceus* | 0.924 | 0.734 | 0.991 | 0.744 | 0.897 | 0.722 | 0.9 | 0.737 | 0.909 | 0.405 |
| *Rhytipterna simplex* | 0.85 | 0.633 | 0.977 | 0.634 | 0.891 | 0.609 | 0.769 | 0.658 | 0.764 | 0.487 |
| *Sclerurus caudacutus* | 0.695 | 0.71 | 0.694 | 0.762 | 0.577 | 0.691 | 0.689 | 0.69 | 0.821 | 0.695 |
| *Thamnomanes caesius* | 0.921 | 0.771 | 0.888 | 0.779 | 0.961 | 0.737 | 0.897 | 0.786 | 0.941 | 0.781 |
| *Thamnophilus palliatus* | 0.806 | 0.567 | 0.758 | 0.594 | 0.802 | 0.544 | 0.815 | 0.528 | 0.846 | 0.604 |
| *Tolmomyias poliocephalus* | 0.885 | 0.635 | 0.981 | 0.606 | 0.909 | 0.65 | 0.814 | 0.65 | 0.834 | 0.319 |
| *Trogon collaris* | 0.95 | 0.56 | 1,000 | 0.595 | 1,000 | 0.559 | 0.99 | 0.526 | 0.81 | 0.275 |
| *Turdus fumigatus* | 0.732 | 0.689 | 0.61 | 0.67 | 0.792 | 0.732 | 0.773 | 0.699 | 0.753 | 0.655 |
| *Xenops minutus* | 0.937 | 0.685 | 0.985 | 0.67 | 0.946 | 0.683 | 0.93 | 0.683 | 0.887 | 0.704 |

The values of the Area Under the Curve (AUC) values for each Amazonian and Atlantic Forest population pair of each species obtained from different methods.

AM = Amazon; MA = Atlantic Forest; ENS = Ensemble; RDF = Random Forest; SVM = Support Vector Machine; GAU = Gaussian; GLM = General Linear Models.

## Niche comparison between birds with disjunct distribution

The first two axes of PCA-env accounted for 70% of the environmental variation in the studied areas. It can be drawn from Table 3 that Amazonian and Atlantic populations have only small Grinellian niche overlap (Schoener's D mean = 0.12; SD = 0.09; range = 0–0.37). *Antrostomus sericocaudatus* and *Cercomacroides laeta* populations' niche did not overlap at all (D = 0) (Fig

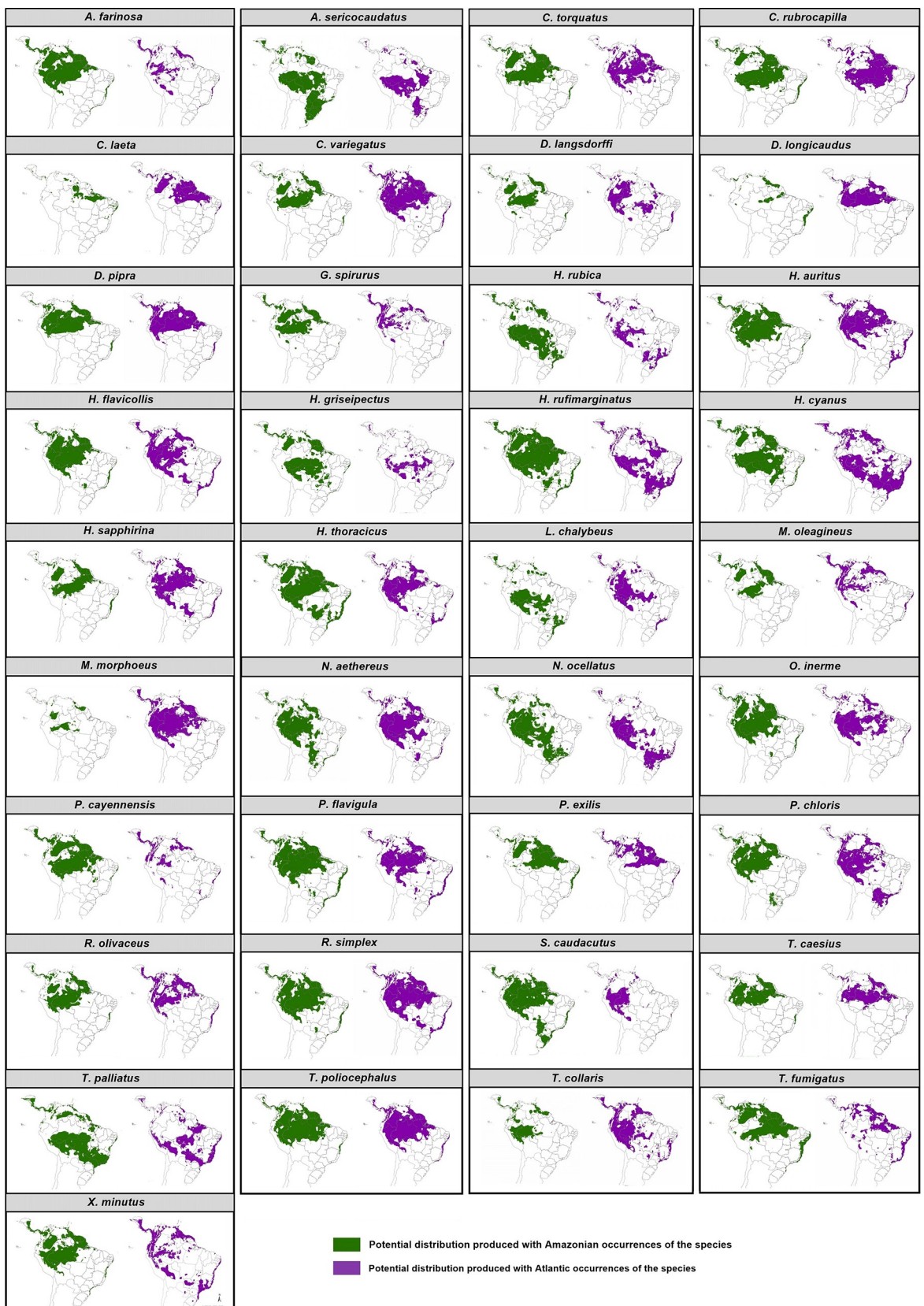

**Fig 2.** Model ensemble of potential distributions produced from occurrence records in both Amazon (green) and the Atlantic Forest (purple) for each species evaluated in the present work.

3). Even though species had low niche overlap values, niche similarity tests indicate that 54% (n = 20) of the species have a more similar niche than would be expected just by chance (p < 0.05).

Stability between Amazonian and Atlantic niches was generally high (mean = 0.81; SD = 0.27). Only 13.5% (n = 5) of the species showed either no or low niche stability (<0.5). Expansion in the Atlantic populations' niche compared to Amazonians was only detected in 21% (n = 8) of the species. In general, populations in the Atlantic Forest only filled partially the niche of Amazonian populations (mean = 0.39; SD = 0.27). Both *A. sericocaudatus* and *C. laeta* that had no niche overlap (D = 0) showed complete expansion and unfilling, and no stability. On the other hand, niches of Atlantic populations of eight species [*Celeus torquatus* (Boddaert, 1783), *Hemitriccus griseipectus* (Snethlage, 1907), *Mionectes oleagineus* (Lichtenstein, 1823), *Monasa morphoeus*, *Rhynchocyclus olivaceus* (Temminck, 1820), *Thamnophilus palliatus* (Lichtenstein, 1823), *Trogon collaris* Vieillot, 1817, and *Xenops minutus* (Sparrman, 1788)] were only a subset of the larger Amazonian niches (no expansion, and complete stability).

## Discussion

Our results indicate that bird populations that have disjunct distribution in the Amazon and Atlantic Forest show signs of Grinellian niche divergence, mainly supported by the low niche overlap among populations of the same species. Although, underlying processes of niche conservatism seemingly constrain niche evolution in these species because for nearly half of the studied species, observed niche overlap—although small—tended to be higher than what would just be expected by chance (similarity test results). Results from the ecological niche models also confirm that the dry diagonal prevents genetic flow between these two forests, as suitable areas almost always fall within the distribution of current forested regions.

[24] reviewed previous tests of niche conservatism in a temporal context, where he found that most of them did not show considerable niche divergence in the time frame examined here (Pleistocene). Our results represent one of the few examples where niche divergence can occur under the such short evolutionary time. One primary mechanism is the lack of gene flow between the populations—supported by phylogeographic studies [75–79] and further inferred by our ecological niche models—that prevent swamping adaptations to the climatic regime characteristics of each forest [26, 80]. Indeed, Atlantic Forest presents more climatic variation and lower temperatures and a smaller volume of precipitation than Amazon [5]. As observed in Fig 2, species niche centroid changes tend to follow the changes in the environmental centroid available in the accessible region for the populations.

When comparing the predictive capacity of the ecological niche models built with Amazonian records, we observed that, for most species, predicted areas agree with the current pattern of occurrence observed in the Atlantic Forest. The same is true for the models of the Atlantic Forest population. These results support a general species niche conservatism of forest habitats constantly recreated by either population, even when their specific niches do not overlap. Niche conservatism—as a process—isolates the populations between Amazon and Atlantic Forest because it besets species adaptation to the conditions found at the dry diagonal.

Atlantic populations' niche showed in general high unfilling, small expansion, and high stability, taking the niches of Amazonian populations as a reference. In other words, niches of the Atlantic Forest populations resemble a subset of that in Amazonian populations. These results support previous genetic evidence of an Amazonian origin of these species [81], which,

**Table 3. Result of niche overlap between populations of Amazonian and Atlantic forest birds, with the result of similarity, expansion, stability and unfilling test.**

| Species | D | Similarity | Expansion | Stability | Unfilling |
|---|---|---|---|---|---|
| *Amazona farinosa* | 0.084 | 0.059 | 0.011 | 0.989 | 0.096 |
| *Antrostomus sericocaudatus* | 0.000 | 1.000 | 1.000 | 0.000 | 1.000 |
| *Campylorhynchus turdinus* | 0.110 | 0.158 | 0.001 | 0.999 | 0.229 |
| *Celeus torquatus* | 0.069 | 0.020 | 0.000 | 1.000 | 0.669 |
| *Ceratopipra rubrocapilla* | 0.196 | 0.109 | 0.554 | 0.446 | 0.562 |
| *Cercomacroides laeta* | 0.000 | 1.000 | 1.000 | 0.000 | 1.000 |
| *Crypturellus variegatus* | 0.232 | 0.030 | 0.056 | 0.944 | 0.104 |
| *Discosura langsdorffi* | 0.047 | 0.069 | 0.226 | 0.774 | 0.667 |
| *Discosura longicaudus* | 0.338 | 0.010 | 0.286 | 0.714 | 0.095 |
| *Dixiphia pipra* | 0.033 | 0.010 | 0.145 | 0.855 | 0.136 |
| *Glyphorhynchus spirurus* | 0.026 | 0.020 | 0.002 | 0.998 | 0.209 |
| *Habia rubica* | 0.308 | 0.079 | 0.014 | 0.986 | 0.399 |
| *Heliothryx auritus* | 0.070 | 0.020 | 0.027 | 0.973 | 0.278 |
| *Hemithraupis flavicollis* | 0.069 | 0.079 | 0.043 | 0.957 | 0.106 |
| *Hemitriccus griseipectus* | 0.095 | 0.079 | 0.000 | 1.000 | 0.460 |
| *Herpsilochmus rufimarginatus* | 0.203 | 0.010 | 0.041 | 0.959 | 0.213 |
| *Hylocharis cyanus* | 0.376 | 0.010 | 0.064 | 0.936 | 0.220 |
| *Hylocharis sapphirina* | 0.115 | 0.059 | 0.285 | 0.715 | 0.565 |
| *Hylophilus thoracicus* | 0.105 | 0.089 | 0.478 | 0.522 | 0.704 |
| *Lophonis chalybeus* | 0.094 | 0.594 | 0.743 | 0.257 | 0.822 |
| *Mionectes oleagineus* | 0.063 | 0.238 | 0.000 | 1.000 | 0.213 |
| *Monasa morphoeus* | 0.052 | 0.010 | 0.000 | 1.000 | 0.424 |
| *Nyctibius aethereus* | 0.034 | 0.109 | 0.615 | 0.385 | 0.697 |
| *Nyctiphrynus ocellatus* | 0.275 | 0.079 | 0.007 | 0.993 | 0.599 |
| *Ornithion inerme* | 0.113 | 0.040 | 0.003 | 0.997 | 0.282 |
| *Panyptila cayennensis* | 0.075 | 0.030 | 0.071 | 0.929 | 0.324 |
| *Piculus flavigula* | 0.103 | 0.030 | 0.449 | 0.551 | 0.451 |
| *Picumnus exilis* | 0.226 | 0.010 | 0.254 | 0.746 | 0.201 |
| *Piprites chloris* | 0.115 | 0.069 | 0.001 | 0.999 | 0.812 |
| *Rhynchocyclus olivaceus* | 0.037 | 0.010 | 0.000 | 1.000 | 0.179 |
| *Rhytipterna simpex* | 0.246 | 0.010 | 0.093 | 0.907 | 0.072 |
| *Sclerurus caudacutus* | 0.116 | 0.040 | 0.095 | 0.905 | 0.155 |
| *Thamnomanes caesius* | 0.054 | 0.010 | 0.323 | 0.677 | 0.674 |
| *Thamnophilus palliatus* | 0.169 | 0.396 | 0.000 | 1.000 | 0.274 |
| *Tolmomyias poliocephalus* | 0.070 | 0.030 | 0.027 | 0.973 | 0.322 |
| *Trogon collaris* | 0.006 | 0.802 | 0.000 | 1.000 | 0.667 |
| *Turdus fumigatus* | 0.207 | 0.010 | 0.035 | 0.965 | 0.171 |
| *Xenops minutus* | 0.131 | 0.010 | 0.000 | 1.000 | 0.087 |

coupled with the process of niche conservatism, would explain the observed pattern—although some species defy this general interpretation (e.g., *A. sericocaudatus* and *C. laeta*).

As previously pointed out by several authors (e.g., [82]), to confirm or not the presence of niche conservatism is not a fundamental approach (although it should surely be tested, see [83]) as to examine the possible consequences of niche conservatism as an ecological and evolutionary process. [26] further explored this topic and proposed that there is a conceptual misinterpretation that niche conservatism presumes the propensity of species to retain the ancestral niche; instead, species would retain their current niche. They called it the

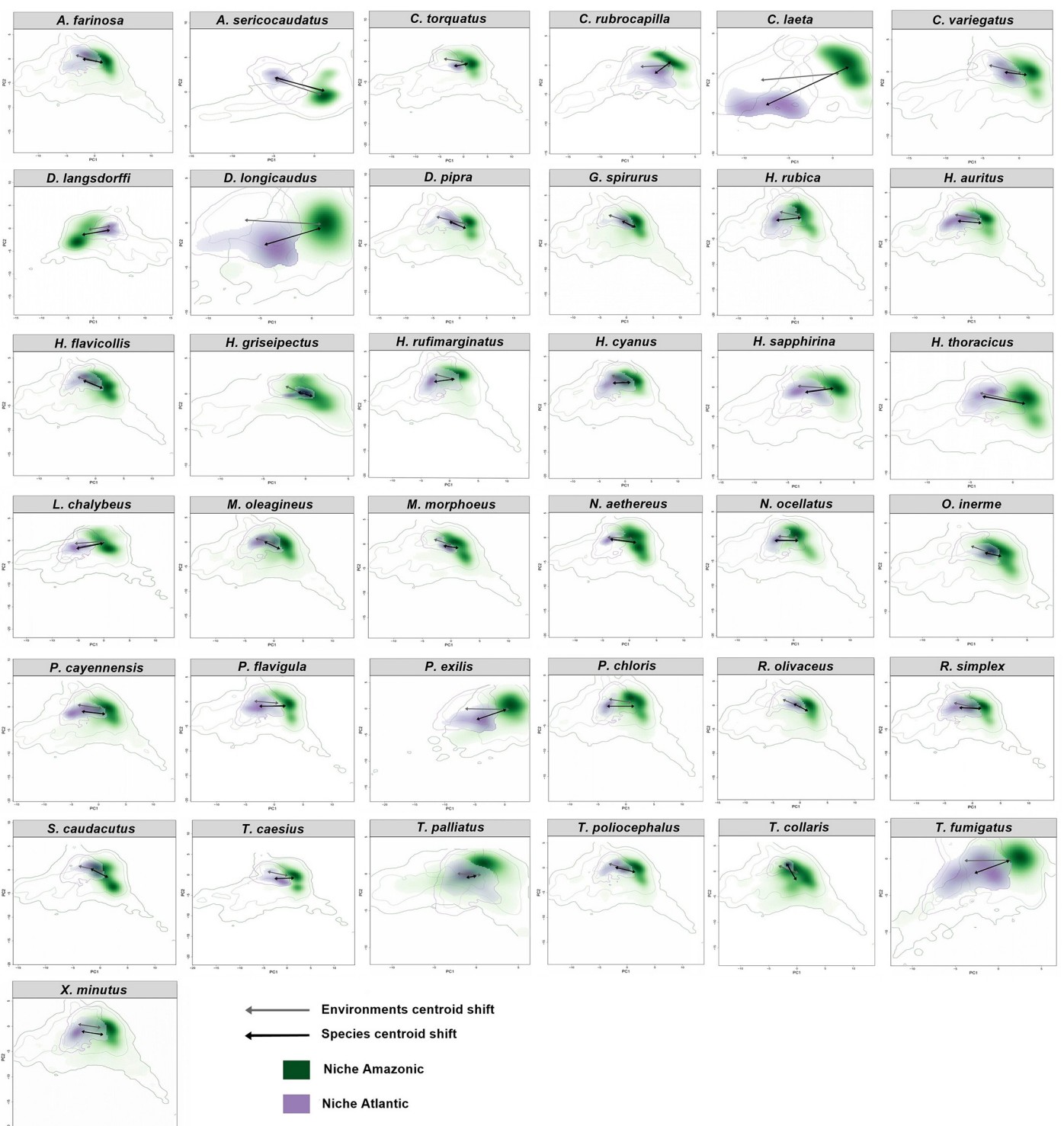

**Fig 3. Niche overlap for each Amazonia—Atlantic forest population pairs for all 38 avian species in this study.** Grey and black arrows indicate environmental and niche centroid shift respectively. Solid and Dashed lines represent 100% and 50% of the available environment (background) for each species, respectively. Antrostomus sericocaudatus's and Cercomacroides laeta's niches do not overlap while for the other species there was only partial niche overlap.

instantaneous niche retention, which is a key concept because, when geographic distance also reflects environmental distance (as in this context), the lack of gene flow associated with divergent natural selection would lead populations to track its instantaneous niche [26]. Therefore, the niche would rapidly evolve as a resulting process of niche conservatism.

These differences could be already driving speciation. For instance, phylogenetic studies indicate that some of the species in our study (such as *A. sericocaudatus*, *C. laeta*, *G. spirurus*, *H. rubica*, and *X. minutus*) are evolutionarily independent units with recognized subspecies in both biomes [75–79]. For instance, *Glyphorynchus spirurus* populations even have significantly different morphological and vocalization patterns [76]. Molecular clock techniques confirm that some of these populations seem to have diverged during the Pleistocene (e.g., *C. laeta*, *G. spirurus*, and *H. rubica*), although for some divergence may have happened before, during the Pliocene (e.g., *X. minutus*) [76–79]. Phylogenetic divergence during Pleistocene has also been observed in primates [84–86], snakes [87], rodents, and marsupials [21, 88]. The diversification of these taxa is consistent with the cycles of isolation of rainforests due to the expansion of savannas during the Pleistocene [1, 4, 21, 84–86, 88–90], supporting this mechanism as an essential current driver of diversification in the Amazon and Atlantic Forest.

Still, it is crucial to bear in mind that the observed niche divergence is not only a result of the most recent isolation of the two forests but likely to be a product of the long process of isolation and recurrent formation of secondary contact zones following the climatic cycles of the quaternary. Accordingly, we advise some caution in inferring the exact time of niche evolution here. Also, as pointed by [91], if both the lack of gene flow (by allopatry or the development of reproductive isolation) and the divergent selection are not stable through time, the role of ecological speciation in driving diversification in the region will not sustain.

## Conclusion

We observed low niche overlap among disjunct populations of the same species that inhabit the Amazon and the Atlantic Forest. However, our results suggest that in 53% of the examined species, the low niche overlap is still higher than predicted under a null model. In general, Grinnellian ecological niches of the population in the Atlantic Forest resemble, to a certain extent, a subset of that of the Amazonian population. However, it is worth noting that some remarkable niche expansions occurred in Atlantic Forest populations. While we have not observed much overlap among the studied species populations, ecological niche models generated with occurrence records of populations from one biome usually recovered the general distribution of populations present on the other. These results lead us to conclude that niche conservatism, while present in many species, was not enough to avoid ecological differentiation among species' Grinnellian ecological niches. In sum, our results support the role of climatic changes that happened at the end of the Pleistocene—that isolated Amazon and Atlantic Forest—as driving ecological differences among the same species populations, and it is also a key mechanism of ongoing diversification in both regions.

## Supporting information

**S1 Data.**
(TXT)

**S2 Data.**
(TXT)

**S1 Fig.**
(TIF)

## Acknowledgments

We thank Prof. Dr. Paulo De Marco and Dr. Santiago Velazco for their support and help with niche modeling; and all citizen scientists that shared their species occurrence data in online databases.

## Author Contributions

**Conceptualization:** Bruno Vilela, Daniel Paiva Silva, Pablo Vieira Cerqueira, Gabriela Silva Ribeiro Gonçalves.

**Formal analysis:** Ricardo Ribeiro da Silva, Bruno Vilela, André Felipe Alves de Andrade.

**Investigation:** Ricardo Ribeiro da Silva, Daniel Paiva Silva.

**Methodology:** Ricardo Ribeiro da Silva, Daniel Paiva Silva.

**Project administration:** Pablo Vieira Cerqueira, Marcos Pérsio Dantas Santos.

**Resources:** Pablo Vieira Cerqueira, Gabriela Silva Ribeiro Gonçalves.

**Supervision:** Daniel Paiva Silva, Marcos Pérsio Dantas Santos.

**Visualization:** Marcos Pérsio Dantas Santos.

**Writing – original draft:** Ricardo Ribeiro da Silva.

**Writing – review & editing:** Ricardo Ribeiro da Silva, Bruno Vilela, Daniel Paiva Silva, André Felipe Alves de Andrade, Pablo Vieira Cerqueira, Gabriela Silva Ribeiro Gonçalves, Marcos Pérsio Dantas Santos.

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
