## [Decision Letter · Decision Letter 0]

4 May 2020

PONE-D-20-05030

The role of ecological niche evolution on speciation patterns of birds  distinctly distributed between the Amazonia and Atlantic Rainforests

PLOS ONE

Dear Mr. Silva,

Thank you for submitting your manuscript to PLOS ONE. After careful consideration, we feel that it has merit but does not fully meet PLOS ONE’s publication criteria as it currently stands. Therefore, we invite you to submit a revised version of the manuscript that addresses the points raised during the review process.

The manuscript is very interesting. I received one review and the referee shared this opinion, but found some major aspects that require reconsideration, such foundation of hypotheses, poorly described methods and sometimes a speculative discussion. Moreover, the text sections in part need better connection and the language might need revision by a native speaker. For detailed comments please see below. I agree with the critics raised by the referee why the decision was 'major revision'.

We would appreciate receiving your revised manuscript by Jun 18 2020 11:59PM. To enhance the reproducibility of your results, we recommend that if applicable you deposit your laboratory protocols in protocols.io, where a protocol can be assigned its own identifier (DOI) such that it can be cited independently in the future. For instructions see: http://journals.plos.org/plosone/s/submission-guidelines#loc-laboratory-protocols

We look forward to receiving your revised manuscript.

Kind regards,

Stefan Lötters

Academic Editor

PLOS ONE

"MPDS were supported by CNPq research productivity fellowships (#308403/2017‐ 7). PVC was supported by a CAPES (the Brazilian Higher Education Training Program) doctoral fellowship (#1537057)."

Please remove any funding-related text from the manuscript and let us know how you would like to update your Funding Statement. Currently, your Funding Statement reads as follows: "No"

4.  We note that Figures 1-3 in your submission contain map images which may be copyrighted. All PLOS content is published under the Creative Commons Attribution License (CC BY 4.0), which means that the manuscript, images, and Supporting Information files will be freely available online, and any third party is permitted to access, download, copy, distribute, and use these materials in any way, even commercially, with proper attribution. For these reasons, we cannot publish previously copyrighted maps or satellite images created using proprietary data, such as Google software (Google Maps, Street View, and Earth). For more information, see our copyright guidelines: http://journals.plos.org/plosone/s/licenses-and-copyright.

1.    You may seek permission from the original copyright holder of Figures 1-3 to publish the content specifically under the CC BY 4.0 license. 

Reviewers' comments:

Reviewer's Responses to Questions

**Comments to the Author**

1. Is the manuscript technically sound, and do the data support the conclusions?

Reviewer #1: Partly

2. Has the statistical analysis been performed appropriately and rigorously? 

Reviewer #1: Yes

3. Have the authors made all data underlying the findings in their manuscript fully available?

Reviewer #1: Yes

4. Is the manuscript presented in an intelligible fashion and written in standard English?

Reviewer #1: No

5. Review Comments to the Author

Reviewer #1: This manuscript investigates whether birds with disjunct distribution in the Amazon and Atlantic forests exhibit either niche conservatism or niche divergence. Authors gathered occurrence data of 37 bird species and built ecological niche models to infer if there is niche overlap between isolated populations in Amazon and Atlantic forests. Despite it has an interesting study question, the manuscript missed foundations of hypotheses. Methods were properly applied. However, authors should consider include Amazon and Atlantic forest in a single model (see below). Sometimes I found discussion very speculative with absence of a properly rejection or acceptance of presented hypotheses in the introduction. The text missed cohesion among subsections. The language needs work. The paper needs a major English language revision by a native speaker. Below, I outlined specific comments:

INTRODUCTION

- I missed a paragraph presenting theoretical basis of niche evolution. Authors gave much focus on Amazon and Atlantic forest disjunction.

- Line 36: you missed Pleistocene.

- Line 45: something is missing between population and species.

- Line 52: authors said “This dynamic may have had some effect on species' niche”. But “some” is vague. You should be more specific.

- Lines 56-58: who said this? Include the reference.

- The arguments presented in introduction did not allow to comprehend why authors evaluated niche overlap of disjunct populations of birds in Amazon and Atlantic forests. Why does check for niche overlap matter? I recommend authors do a better job in reviewing theoretical basis of niche evolution. Then, they will be able indicate why is important check niche overlap for these birds.

- Lines 87-91: Here authors tried to present the study hypotheses (I could not understand the second hypothesis, sentence need to be rewrite). However, it was poor explored. Authors should do a better job in presenting theories that support such hypotheses. In addition, they must present expectations of their hypotheses according to methods they area applying. Then, the readers will be able tom comprehend which hypothesis will be accepted.

MATERIAL AND METHODS

I found methods well written. Sometimes it seems that was wrote for a different person that produced the manuscript.

- Why did the authors only use temperature and precipitation data from Bioclim to infer the niche of species? Why do not include other variables, such as vegetation cover (e.g. EVI: https://www.earthenv.org/), solar radiation, wind speed, water vapor pressure (they are available at the new worldclim version https://www.worldclim.org/data/worldclim21.html), etc?

RESULTS

- why did model Amazon and Atlantic forest separately? As there is evidence of niche conservatism due to a recent divergence, it is better to build a single model for both biomes. Then they can split Amazon and Atlantic Forest after build the models and infer niche overlap. Otherwise, if they decide to maintain their original procedure a convincing justification must be presented.

- I am thinking if the niche overlap result observed in most species would not be biased by modeling Amazon and Atlantic forest separately. Ecological niche models for most species recovered the distribution in Amazon when modelling Atlantic Forest, or vice-versa (as seen in figure 2). However, if the authors aim to infer niche overlap between disjunct populations in these two forests, the niche inferred by Atlantic Forest populations must not include areas in Amazon, because species cannot disperse into that, despite similarities in temperature and precipitation. The opposite happens for Amazon populations.

- I missed a figure and/or table that summarize/combine the niche overlap of all species. I could not find the D values of each species comparison. Maybe include a graph with D values for each species. It will provide an easier way to observe a general pattern in your results.

DISCUSSION

- I found some parts of discussion quite confuse and not linked to the results. In lines 267-271, authors said that found niche overlap, but in results they said that species presented low niche overlap (Schoener's D values below 0.4) (lines 248-249). I cannot follow this, it is very confused.

- lines 280-282: how many was predicted? in how many species?

- lines 287-289: again, different from results in lines 248-249.

- lines 316-326: It is quite speculative. Authors can analyze the genetic data available for some species to check this. Otherwise, they must consider remove this from the text.

- Lines 327: it is also in Pliocene.

- Lines 349-351: I did not understand. If they have subspecies one would expect exactly no niche overlap.

- lines 395: You did not measure phylogenetic diversity.

6. PLOS authors have the option to publish the peer review history of their article (what does this mean?). If published, this will include your full peer review and any attached files.

Reviewer #1: No

---

## [Author Response · Author response to Decision Letter 0]

12 Jul 2020

ANSWERS TO THE REVIEWERS

We made the necessary adjustments within the style requirements of PLOS ONE. 

We inserted the present ORCID ID number 0000-0003-4979-5302 and validated in the Editorial Manager. 

"MPDS were supported by CNPq research productivity fellowships (#308403/2017‐ 7). PVC was supported by a CAPES (the Brazilian Higher Education Training Program) doctoral fellowship (#1537057)." We note that you have provided funding information that is not currently declared in your Funding Statement. However, funding information should not appear in the Acknowledgments section or other areas of your manuscript. We will only publish funding information present in the Funding Statement section of the online submission form.

Please remove any funding-related text from the manuscript and let us know how you would like to update your Funding Statement. Currently, your Funding Statement reads as follows: "No"

The information presented in the acknowledgments refers to a scholarship, reason for acknowledgement from the authors. The present study did not have any financial support.

4. We note that Figures 1-3 in your submission contain map images which may be copyrighted. All PLOS content is published under the Creative Commons Attribution License (CC BY 4.0), which means that the manuscript, images, and Supporting Information files will be freely available online, and any third party is permitted to access, download, copy, distribute, and use these materials in any way, even commercially, with proper attribution. For these reasons, we cannot publish previously copyrighted maps or satellite images created using proprietary data, such as Google software (Google Maps, Street View, and Earth). For more information, see our copyright guidelines: http://journals.plos.org/plosone/s/licenses-and-copyright.

We believe that it is important to emphasize that the map figures presented in this manuscript were built using ArcGis software using shapefile files that contain polygons that make up a world map of political boundaries. We did not use satellite images or figures protected by copyright for the construction of the map figures. We did all figures, and we believe there is no copyright issues in our manuscript.

We made the necessary adjustments.

Reviewer:

Comments to the Author

Reviewer #1

Q#1: This manuscript investigates whether birds with disjunct distribution in the Amazon and Atlantic forests exhibit either niche conservatism or niche divergence. Authors gathered occurrence data of 37 bird species and built ecological niche models to infer if there is niche overlap between isolated populations in Amazon and Atlantic forests. Despite it has an interesting study question, the manuscript missed foundations of hypotheses. Methods were properly applied. However, authors should consider include Amazon and Atlantic forest in a single model (see below). Sometimes I found discussion very speculative with absence of a properly rejection or acceptance of presented hypotheses in the introduction. The text missed cohesion among subsections. The language needs work. The paper needs a major English language revision by a native speaker. 

We appreciate all the reviewer effort in reviewing our manuscript. As mentioned above, we did substantial changes to improve the writing and the theoretical foundation of our manuscript. We address each specific suggestion below.

Below, I outlined specific comments:

I missed a paragraph presenting theoretical basis of niche evolution. Authors gave much focus on Amazon and Atlantic forest disjunction.

A#1: We restructured the introduction to include a general introduction of niche evolution and why it is important to our work. Changes are in the second and third paragraphs of the intorduction. 

Q#2: Line 36: you missed Pleistocene.

A#2: As we restructured the text of the introduction, this sentence is no longer in the text. We took care not to make mistakes when citing the geological time and its dating. 

Q#3: Line 45: something is missing between population and species.

Q#3: This sentence is no longer in the text.

Q#4: Line 52: authors said “This dynamic may have had some effect on species' niche”. But “some” is vague. You should be more specific.

Q#4: We agree with the review. The information was vague. We restructured the text and avoided using vague sentences throughout the manuscript.

Q#5: Lines 56-58: who said this? Include the reference.

A#5 We removed this part of the manuscript and restructured this paragraph. 

Q#6: The arguments presented in introduction did not allow to comprehend why authors evaluated niche overlap of disjunct populations of birds in Amazon and Atlantic forests. Why does check for niche overlap matter? I recommend authors do a better job in reviewing theoretical basis of niche evolution. Then, they will be able indicate why is important check niche overlap for these birds.

Q#6: We appreciate the constructive criticism. As mentioned before, we restructured the introduction to make our arguments clearer. We believe that our introduction do a much better job in presenting the proper arguments and justifications for our work. 

Q#7: Lines 87-91: Here authors tried to present the study hypotheses (I could not understand the second hypothesis, sentence need to be rewrite). However, it was poor explored. Authors should do a better job in presenting theories that support such hypotheses. In addition, they must present expectations of their hypotheses according to methods they area applying. Then, the readers will be able tom comprehend which hypothesis will be accepted.

A#7: We agree and reformulate the wording for a better understanding of the hypothesis of the article. In the last paragraph of the introduction, we present the aims of the paper as: 

“We intend to test whether the 37 species of birds with disjunct distribution in the Amazon and the Atlantic Forest show signs of rapid evolution in the Grinnelian niche, using two approaches: (I) we first tested whether ecological niche models constructed with records of the occurrence of a population could predict the distribution of the other; (II) and we also examined the niche overlap between populations using a recently developed structure, which considers the differences in the spatially available environmental conditions in each region - which, if not taken into account, could mask the real evolution of the niche.”

Q#8: Why did the authors only use temperature and precipitation data from Bioclim to infer the niche of species? Why do not include other variables, such as vegetation cover (e.g. EVI: https://www.earthenv.org/), solar radiation, wind speed, water vapor pressure (they are available at the new worldclim version https://www.worldclim.org/data/worldclim21.html), etc?

A#8:. We avoid using the EVI because it is a metric that has great seasonal / temporal variation and as we are testing a possible evolution in the niche, our time scale is broader. In the same way, land use cover metrics from EarthEnv do not reflect the species’ historical niche. Otherwise variables such as wind speed and water vapor pressure are interesting variables for migratory birds (Klaassen M 1996. The Journal of experimental biology. issn: 1477-9145; Carmi et al 1992. The Auk. doi: 10.2307/4088195; Alerstam T. Ornis Scandinavica. doi: 10.2307/3676347), which does not cover our species.

Q#9: why did model Amazon and Atlantic forest separately? As there is evidence of niche conservatism due to a recent divergence, it is better to build a single model for both biomes. Then they can split Amazon and Atlantic Forest after build the models and infer niche overlap. Otherwise, if they decide to maintain their original procedure a convincing justification must be presented.

A#9: We made some changes to the Ecological Niche Model in the Methods section (L133-142) that we hope makes this point clearer. We built separate niche models to test for a niche evolution / differentiation between isolated populations of the same species. This strategy is similar to other key studies in the field (Peterson & Holt 2003. Ecology Letters. doi: 10.1046/j.1461-0248.2003.00502.x; Dowell et al. 2016.Royal Society Open Science. doi: 10.1098/rsos.150619; Hällfors et al. 2015.Ecological Applications.doi: 10.1890/15-0926.1). By adjusting the model for one population and evaluating the effectiveness of this model when predicting the other population, we obtain an estimate of niche change, since these models characterize the niche of the separated populations of the species. If the reciprocal models fail to predict one another, it is an indication of potential niche evolution. Since SDMs are static methods, by applying such methods it is possible to better understand potential niche evolution of the modeled species.

Q#10: I am thinking if the niche overlap result observed in most species would not be biased by modeling Amazon and Atlantic forest separately. Ecological niche models for most species recovered the distribution in Amazon when modelling Atlantic Forest, or vice-versa (as seen in figure 2). However, if the authors aim to infer niche overlap between disjunct populations in these two forests, the niche inferred by Atlantic Forest populations must not include areas in Amazon, because species cannot disperse into that, despite similarities in temperature and precipitation. The opposite happens for Amazon populations.

A#10: We agree that, while fitting the models, we must avoid sampling areas from the other regions, as species never reached that region. Therefore, we took the precautions to avoid sampling pseudo-absences/background data in the opposite region by restricting sampling with a buffer around occurrences, as failing to do so would artificially create a sampling bias (Phillips 2008. Ecography. doi: 10.1111/j.0906-7590.2008.5378.x). Furthermore, we were interested in how well a model fitted on occurrence data from one region was able to accurately predict species’ occurrence at the other region. We believe this procedure was not clear in the methods section and expect that this is made clearer with the recent changes (L133-142).

Q#11: I missed a figure and/or table that summarize/combine the niche overlap of all species. I could not find the D values of each species comparison. Maybe include a graph with D values for each species. It will provide an easier way to observe a general pattern in your results.

A#11: We reviewed and inserted table 3 with the values of the niche overlap for all species for better understanding of the manuscript.

Q#12: I found some parts of discussion quite confuse and not linked to the results. In lines 267-271, authors said that found niche overlap, but in results they said that species presented low niche overlap (Schoener's D values below 0.4) (lines 248-249). I cannot follow this, it is very confused.

A#12: We agreed and restructured a sentence to better understand the text.

Q#13: lines 280-282: how many was predicted? in how many species?

A#13: Prediction occurred in 18 species in both the north and south of the Atlantic Forest, representing 49% of the species in the model. Currently, the text is at lines 268-270.

Q#14: lines 287-289: again, different from results in lines 248-249.

A#14: We agreed and restructured a sentence to better understand the text. Our results show that Amazonian and Atlantic populations have only small Grinellian niche overlap (Schoener's D mean = 0.12; SD = 0.09; range = 0 – 0.37). Antrostomus sericocaudatus and Cercomacroides laeta populations’ niche did not overlap at all (D = 0). Even though species had low niche overlap values, niche similarity tests indicate that 54% (n = 20) of the species have a more similar niche than would be expected just by chance (p < 0,05). The data of the niche overlap are present in Figure 3 of the manuscript.

Q#15: lines 316-326: It is quite speculative. Authors can analyze the genetic data available for some species to check this. Otherwise, they must consider remove this from the text.

A#15: We agreed and removed it from the manuscript.

Q#16: Lines 327: it is also in Pliocene.

A#16: We agreed and changed in the text according to suggestion from the reviewer.

Q#17: Lines 349-351: I did not understand. If they have subspecies one would expect exactly no niche overlap.

A#17: We agreed and restructured a sentence to better understand the text. Our analyzes corroborate studies that indicate that some of the species in our study (such as Antrostomus sericocaudatus, Cercomacroides laeta, Glyphorynchus spirurus, Habia rubica and Xenops minutus) are evolutionarily independent units with recognized subspecies in both biomes (Marks et al. 2002. Molecular Phylogenetics and Evolution. doi: 10.1016/S1055-7903(02)00233-6; Fernandes et al. 2013. Molecular Phylogenetics and Evolution. doi: 10.1016/j.ympev.2012.09.033; Tello et al. 2014. Zoological Journal of the Linnean Society. doi: 10.1111/zoj.12116; Harvey and Brumfield 2015. Molecular Phylogenetics and Evolution. doi: 10.1016/j.ympev.2015.04.018; Lavinia et al. 2015. Molecular Phylogenetics and Evolution. doi: 10.1016/j.ympev.2015.04.018).” 

Q#18: lines 395: You did not measure phylogenetic diversity.

A#18: We agreed and changed in the text according to suggestion from the reviewer. We do not measure phylogenetic diversity.

---

## [Decision Letter · Decision Letter 1]

13 Aug 2020

PONE-D-20-05030R1

The role of ecological niche evolution on speciation patterns of birds  distinctly distributed between the Amazonia and Atlantic Rainforests

PLOS ONE

Dear Dr. Silva,

Thank you for submitting your manuscript to PLOS ONE. After careful consideration, we feel that it has merit but does not fully meet PLOS ONE’s publication criteria as it currently stands. Therefore, we invite you to submit a revised version of the manuscript that addresses the points raised during the review process.

Thanks for improving your paper. The (same) referee is satisfied, so am I. there are a few minor things left, as raised by the referee. Please consider them accordingly and submit your revised manuscript by Sep 27 2020 11:59PM. If you will need more time than this to complete your revisions, please reply to this message or contact the journal office at plosone@plos.org. Please include the following items when submitting your revised manuscript:

We look forward to receiving your revised manuscript.

Kind regards,

Stefan Lötters

Academic Editor

PLOS ONE

Reviewers' comments:

Reviewer's Responses to Questions

**Comments to the Author**

1. If the authors have adequately addressed your comments raised in a previous round of review and you feel that this manuscript is now acceptable for publication, you may indicate that here to bypass the “Comments to the Author” section, enter your conflict of interest statement in the “Confidential to Editor” section, and submit your "Accept" recommendation.

Reviewer #1: All comments have been addressed

2. Is the manuscript technically sound, and do the data support the conclusions?

Reviewer #1: Yes

3. Has the statistical analysis been performed appropriately and rigorously? 

Reviewer #1: Yes

4. Have the authors made all data underlying the findings in their manuscript fully available?

Reviewer #1: Yes

5. Is the manuscript presented in an intelligible fashion and written in standard English?

Reviewer #1: Yes

6. Review Comments to the Author

Reviewer #1: Authors made a very good job in reviewing the manuscript. This new version is substantially improved. I am satisfied with modifications and author’s responses. I have some minor comments:

TITLE: change the word "speciation" to "diversification" as most of isolated populations on both biomes are not ranked as distinct species.

Lines 26-28: This statement was proposed by classical papers, instead these new ones. Please correct the citation including Haffer (1969) and Brown & Ab’Saber (1979).

Lines 194-197: Why do the authors choose these thresholds? It is better use commonly used values from previous studies, than arbitrary values.

Line 233: correct is 0.05.

Lines 247-248: Actually, your results indicate both niche divergence and conservatism. You should put in evidence both scenarios, as evidenced in your results.

Lines 257-258: what do you mean with "considered evolutionary time"? We might expect recent splits for most co-distributed species in both biomes, due to lack of phenotypic divergence.

Line 259: which phylogeographic studies?

Lines 301-302: include Ledo and Coli (2017) and Batalha-Filho et al (2013) in the citations, as they properly showed this hypothesis.

References:

Batalha-Filho H, Fjeldså J, Fabre P-H, et al (2013) Connections between the Atlantic and the Amazonian forest avifaunas represent distinct historical events. J Ornithol 154:41–50.

Brown, K.S., Ab’Saber, A.N., 1979. Ice-ages forest refuges and evolution in the Neotropics: correlation of paleoclimatological, geomorphological, and pedological data with modern biological endemism. Paleoclimas 5, 1–30.

Haffer, J., 1969. Speciation in Amazonian forest birds. Science 165, 131–137.

Ledo, RMD, Colli, GR. The historical connections between the Amazon and the Atlantic Forest revisited. J Biogeogr. 2017; 44: 2551– 2563.

7. PLOS authors have the option to publish the peer review history of their article (what does this mean?). If published, this will include your full peer review and any attached files.

Reviewer #1: No

---

## [Author Response · Author response to Decision Letter 1]

19 Aug 2020

ANSWERS TO THE REVIEWERS

Reviewer:

Comments to the Author

Reviewer #1

Q#1: Authors made a very good job in reviewing the manuscript. This new version is substantially improved. I am satisfied with modifications and author’s responses. I have some minor comments:

Thank you for the positive feedback, your comments were fundamental to guide our revision. 

TITLE: change the word "speciation" to "diversification" as most of isolated populations on both biomes are not ranked as distinct species.

Q#1: We changed the word "speciation" to "diversification" in the title.

Q#2: Lines 26-28: This statement was proposed by classical papers, instead these new ones. Please correct the citation including Haffer (1969) and Brown & Ab’Saber (1979).

A#2: Thanks for pointing this out. We have corrected the citation including these articles. 

Q#3: Lines 194-197: Why do the authors choose these thresholds? It is better use commonly used values from previous studies, than arbitrary values. 

Q#3: We changed and standardized the limit value based on previous studies (Manel et al. 1999, 2001, Luck 2002, Stockwell and Peterson 2002, Bailey et al. 2002, Woolf et al. 2002, Liu 2005), considering the value of 0.05 in the lowest category to optimize the percentage of models. 

Q#4: Line 233: correct is 0.05. 

Q#4: We modified the text to the correct value. 

Q#5: Lines 247-248: Actually, your results indicate both niche divergence and conservatism. You should put in evidence both scenarios, as evidenced in your results. 

Q#5: We agree, the entire paragraph show that. But we moved the sentence up to make it clearer. The text now reads as follow:

“Our results indicate that bird populations that have disjunct distribution in the Amazon and Atlantic Forest show signs of Grinellian niche divergence, mainly supported by the low niche overlap among populations of the same species. Although, underlying processes of niche conservatism seemingly constrain niche evolution in these species because for nearly half of the studied species, observed niche overlap — although small — tended to be higher than what would just be expected by chance (similarity test results).”

Q#6: Lines 257-258: what do you mean with "considered evolutionary time"? We might expect recent splits for most co-distributed species in both biomes, due to lack of phenotypic divergence.

A#6 We were referring to the short evolutionary time. We changed the sentence to be more specific and avoid confusion. The sentence now reads as follow:

“Our results represent one of the few examples where niche divergence can occur under the such short evolutionary time.”

Q#7: Line 259: which phylogeographic studies?

Q#7: We updated the text to indicate the proper references of the phylogeographic studies.

Q#8: Lines 301-302: include Ledo and Coli (2017) and Batalha-Filho et al (2013) in the citations, as they properly showed this hypothesis.

Q#8: We added the appropriate citations.

---

## [Editor Report · Decision Letter 2]

24 Aug 2020

The role of ecological niche evolution on diversification  patterns of birds distinctly distributed between the Amazonia and Atlantic Rainforests

PONE-D-20-05030R2

Dear Dr. Silva,

We’re pleased to inform you that your manuscript has been judged scientifically suitable for publication and will be formally accepted for publication once it meets all outstanding technical requirements.

Kind regards,

Stefan Lötters

Academic Editor

PLOS ONE
---

## [Editor Report · Acceptance letter]

3 Sep 2020

PONE-D-20-05030R2 

The role of ecological niche evolution on diversification  patterns of birds distinctly distributed between the Amazonia and Atlantic Rainforests 

Dear Dr. Silva:

I'm pleased to inform you that your manuscript has been deemed suitable for publication in PLOS ONE. Congratulations! Your manuscript is now with our production department. 

Kind regards, 

on behalf of

Prof. Dr. Stefan Lötters 

Academic Editor

PLOS ONE